# POINCARENORM: RETHINKING OVER-SMOOTHING BEYOND DIRICHLET ENERGY

## ABSTRACT

Dirichlet energy is intuitive and commonly used to measure over-smoothing. However, Dirichlet energy can only capture information about the first-order derivative of features. In light of this, we propose a series of node similarity measures which are the energy of higher-order derivatives of features and generalize Dirichlet energy. After we rigorously analyze the property of proposed measures and its application to establish the sharp decay rate of Dirichlet energy under continuous diffusion or discrete random walk which is closely related to the first nonzero eigenvalue of graph Laplacian. Lastly, to address over-smoothing with respect to these measures, we propose a normalization termed PoincareNorm which generalizes PairNorm to control our proposed measures. We consider the semi-supervised node classification task in the scenario without missing features, PoincareNorm outperforms existing normalization methods.

## 1 INTRODUCTION

Graph neural networks(GNNs) have emerged recently and successfully applied to many fields such as protein prediction (Gligorijević et al., 2021) and social recommendation (Fan et al., 2022). Though vanilla graph convolutional network(GCN) (Kipf and Welling, 2017), graph attention network(GAT) (Veličković et al., 2018) and a class of neural networks following the paradigm of message passing type (Gilmer et al., 2017) have achieved great success in many graph-based tasks, they suffer from over-smoothing when stacking layers. Many works theoretically analyze the loss of expressive power for attention-based mechanism (Wu et al., 2023) and graph convolutional network (Oono and Suzuki, 2020)(Cai and Wang, 2020). Works are emerging to address over-smoothing such as residual connection (Li et al., 2019) (Liu et al., 2020) (Chen et al., 2020)(Xu et al., 2018) originated from (He et al., 2016), regularization such as dropout (Papp et al., 2021) (Rong et al., 2020)(Fang et al., 2023). Nguyen et al. (2023) think that positive Ollivier-Ricci curvature will cause over-smoothing and they delete edges with highly positive curvature. From the spectrum view over-smoothing is closely related to the first nonzero eigenvalue of graph Laplacian (Jamadandi et al., 2024) (Giraldo et al., 2023).

Simplified graph convolution(SGC) (Wu et al., 2019) separates transformation and propagation. With propagation only they can stack layers and achieve competing performance with GCN, Wang et al. (2021) further use heat kernel to propagate and Li et al. (2022) consider more propagation kernel. Another mainstream to address over-smoothing is to consider different ordinary differential equations(ODE) and partial differential equations(PDE) to change the dynamic system of GNNs. Rusch et al. (2023b) use gradient gating to control the update of features. Kang et al. (2024) apply fractional time derivative to evolving dynamic system. More equation-based GNNs such as Allen-Cahn equation (Wang et al., 2023), reaction-diffusion equation (Choi et al., 2023). For more equation-based graph neural networks we refer to (Rusch et al., 2022)(Maskey et al., 2023)(Behmanesh et al., 2023)(Thorpe et al., 2022)(Chamberlain et al., 2021)(Xhonneux et al., 2020).

Normalization of features has also been proven successful in graph neural networks. a few works are devoted to address over-smoothing such as PairNorm (Zhao and Akoglu, 2020) and GroupNorm (Zhou et al., 2020). These works keep normalized features away from trivial under the measure of over-smoothing. NodeNorm (Zhou et al., 2021a) think of degradation of graph convolutional network as variance inflammation and normalize over feature dimension for every node similar to layernorm (Ba et al., 2016), GraphNorm (Cai et al., 2021) think that usual instancenorm (Ulyanov et al., 2017)

applied to GNN will cause loss of information due to standard shift, instead they normalize the feature values across all nodes in each graph with a learnable shift.

Many measures are proposed to quantitively measure over-smoothing, among which Dirichlet energy is intuitively and most commonly used (Rusch et al., 2022)(Zhou et al., 2021b), however, Dirichlet energy can only capture information about the first-order derivative of features. Also, Dirichlet energy based normalization PairNorm can not control the energy of higher-order derivatives well. Based on this observation our contributions are as follows

- We propose a series of node similarity measures to measure over-smoothing more finely, this measure generalizes Dirichlet energy. We prove that these node similarity measures satisfy the condition of (Rusch et al., 2023a). Then we theoretically illustrate the relation between proposed node similarity measures and use this relation to build up the decay rate of Dirichlet energy under continuous heat diffusion and general discrete random walk. Results of the decay rate will show that self-loop is a necessary condition for over-smoothing. Also, the decay rate under continuous heat diffusion can give a more concrete analysis of over-smoothing of the work (Wang et al., 2021) and indicate the relation between over-smoothing and the first nonzero eigenvalues of graph Laplacian.

- In light of these measures, We use Poincare inequality to construct normalization termed PoincareNorm to control measures proposed by us, we note that this normalization is a generalization of PairNorm. We consider the semi-supervised node classification task in the scenario without missing features, PoincareNorm outperforms existing normalization methods.

## 2 PRELIMINARY AND NOTATION

We introduce the framework of weighted graphs and calculus on it, this framework is commonly used when studying differential equations on graphs (Grigor'yan et al., 2016)(Sun and Wang, 2022), we make use of this framework and show that attention-based GNNs can be generalized into this framework. In this article all graphs are assmued to be finite and undirected. Let $G = (V, E, \omega, \mu)$ be a weighted graph, where $V = \{1, 2, ..., n\}$ is the set of nodes, $E \subset V \times V$ is the set of edges, $\omega$ is a function on $E$ such that $\omega_{ij} = \omega([i, j]) = \omega([j, i]) > 0$ for $[i, j] \in E$, $\mu$ is a function on $V$ such that $\mu_i = \mu(i) > 0$ for $i \in V$. Fixed $i \in [n]$, $\mathcal{N}_i = \{j \in [n] : \omega_{ij} > 0, j \neq i\}$ denotes node $i$'s neighborhood. Denote $\omega_{max} = \max_{i \in [n], j \in \mathcal{N}_i} \omega_{ij}$, $\omega_{min} = \min_{i \in [n], j \in \mathcal{N}_i} \omega_{ij}$, $\mu_{max} = \max_{i \in [n]} \mu_i$, $\mu_{min} = \min_{i \in [n]} \mu_i$, $|V|_\mu = \sum_{i \in [n]} \mu_i$. Following we define the general gradient and Laplacian of a vectored value function $f$.

**Definition 2.1.** *Let $G = (V, E, \omega, \mu)$ be a weighted graph and $f : V \to \mathbb{R}^d$ is vector valued funtion on the nodes,integration of $f$ on $V$ is defined as*

$$\int_V f d\mu = \sum_{i=1}^n f(i)\mu(i)$$

*The inner product of the gradient of f and the gradient of g is defined as*

$$\int_V \nabla_\mu f \cdot \nabla_\mu g d\mu = \sum_{j \in \mathcal{N}_i} \frac{\omega_{ij}(f(j) - f(i)) \cdot (g(j) - g(i))}{2\mu_i}$$

*Where $\cdot$ is dot product between vectors. General p-norm of a vector $x = (x_1, \ldots, x_k)$ is defined as*

$$\|x\|_p = (\sum_{i=1}^k |x_i|^p)^{\frac{1}{p}}$$

*where $p \geq 1$. p-norm of Gradient of f on V is defined as*

$$\|\nabla_\mu f\|_p(i) = (\sum_{j \in \mathcal{N}_i} \frac{\omega_{ij}\|f(j) - f(i)\|_p^p}{2\mu(i)})^{\frac{1}{p}}$$

*Laplacian of f on V is defined as*

$$\Delta_\mu f(i) = \sum_{j \in \mathcal{N}_i} \frac{\omega_{ij}(f(j) - f(i))}{\mu_i}$$

The virtue of the above definition is that we can integrate by parts. This is a well-known result and we give a proof for completeness. Proof of the following Theorem and all proof needed in this paper will be left in the Appendix.

**Theorem 2.1.** *Let $G = (V, E, \omega, \mu)$ be a weighted graph and $f, g : V \to \mathbb{R}^d$ are vector-valued functions on the nodes, then we have integration by parts*

$$\int_V \Delta_\mu f \cdot g d\mu = - \int_V \nabla_\mu f \cdot \nabla_\mu g d\mu = \int_V f \cdot \Delta_\mu g d\mu$$

Thus we can do calculus on graphs under the above general definition of gradient and Laplacian. Now recall traditionally considered graph Laplacian, $A \in \mathbb{R}^{n \times n}, D = diag(D_i, \ldots, D_n), \tilde{A} = A + I$ and $\tilde{D}$ denotes adjacent matrix, degree matrix, adjacent matrix with self-loop and degree matrix of $\tilde{A}$ respectively. $\tilde{A}_{sym} = \tilde{D}^{-\frac{1}{2}} \tilde{A} \tilde{D}^{-\frac{1}{2}}$ and $\tilde{A}_{rw} = \tilde{D}^{-1} \tilde{A}$ are denoted as symmetric normalization and row normalization of $\tilde{A}$ respectively. When there exist no isolated nodes, we can also define $A_{sym} = D^{-\frac{1}{2}} A D^{-\frac{1}{2}}$ and $A_{rw} = D^{-1} A$. Graph Laplacian with respect to $A, \tilde{A}, \tilde{A}_{sym}, \tilde{A}_{rw}, A_{sym}, A_{rw}$ and are defined as $\Delta_{adj} = A - D, \tilde{\Delta}_{adj} = \tilde{A} - \tilde{D}, \tilde{\Delta}_{sym-adj} = \tilde{A}_{sym} - I, \tilde{\Delta}_{rw-adj} = \tilde{A}_{rw} - I, \Delta_{sym-adj} = A_{sym} - I$ and $\Delta_{rw-adj} = A_{rw} - I$ respectively. We give special cases of general graph Laplacian when $\Delta = \Delta_{adj}, \tilde{\Delta}_{adj}, \Delta_{rw-adj}$ or $\tilde{\Delta}_{rw-adj}$ as follow

$$\begin{cases} \omega_{ij} = 1, \mu_i = 1 & \text{when} \quad \Delta = \Delta_{adj} \\ \omega_{ij} = 1, \mu_i = 1 & \text{when} \quad \Delta = \tilde{\Delta}_{adj} \\ \omega_{ij} = 1, \mu_i = D_i + 1 & \text{when} \quad \Delta = \tilde{\Delta}_{rw-adj} \\ \omega_{ij} = 1, \mu_i = D_i & \text{when} \quad \Delta = \Delta_{rw-adj} \end{cases}$$

From theorem 2.1, $-\Delta_\mu$ is a semi-positive symmetric operator on function space $L^2(V, d\mu) = \{f : V \to \mathbb{R}^d : \int_V \|f\|_2^2 d\mu < \infty\}$. we array its eigenvalues as follows

$$0 = \lambda_0(-\Delta) < \lambda_1(-\Delta_\mu) < \ldots < \lambda_N(-\Delta_\mu)$$

where $N \geq 1$. Denote $\mu_i^1 = \sum_{j \in \mathcal{N}_i} \omega_{ij}, M_{max} = \max_{i \in [n]} \frac{\sum_{j \in \mathcal{N}_i} \omega_{ij}}{\mu_i} = \max_{i \in [n]} \frac{\mu_i^1}{\mu_i}$, then we have an upper bound for $\lambda_N$ as follows

**Theorem 2.2.** *Given a weighted graph $G = (V, E, \omega, \mu)$, then we have upper bound for $\lambda_N$*

$$\lambda_N \leq 2M_{max}$$

If $\forall i \in [n], \sum_{j \in \mathcal{N}_i} \omega_{ij} \leq \mu_i$, then $P_\mu = \Delta_\mu + I$ is called random walk matrix associated with $\Delta_\mu$, we say that $P_\mu$ satisfies self-loop if $\forall i \in [n], \sum_{j \in \mathcal{N}_i} \omega_{ij} < \mu_i$. We have upper bound $\lambda_N \leq 2$. We recall that the layer of attention-based GNNs is as follows

$$X^{l+1} = \phi(P^l X^l W^l)$$

where $\phi$ is the activation function, $W^l$ is learnable matrix and $P^l$ is an aggregation operator in message-passing. $P_{ij}^l$ denotes the i-th row and the j-th column element of $P^n$, usually $P_{ij}^l$ can be expressed as

$$P_{ij}^l = \frac{exp(e_{ij}^l)}{exp(e_{ii}^l) + \sum_{k \in \mathcal{N}_i} exp(e_{ik}^l)}$$

where $e_{ij}^l$ is weight in layer $l$ between node i and j. This aggregation operator coincides with our definition of random matrix if we specify edge weights $\omega_{ij}$ and node weights $\mu$ as follows

$$\omega_{ij} = exp(e_{ij}), \quad \mu_i = exp(e_{ii}^l) + \sum_{k \in \mathcal{N}_i} exp(e_{ik}^l)$$

## 3 UNDERSTANDING OVER-SMOOTHING

In this work, we consider the semi-supervised node classification task on the graph. Every node $i$ in $V$ is given features $X_i$. Only a subset of $V$ are given labels. A training set $V_{train}$ is a subset of $V$ with labels for training, the task aims to predict the labels of the nodes $V \backslash V_{train}$ from features.

## 3.1 MEASURES OF OVER-SMOOTHING

There exists a variety of approaches to quantitatively measure the issue of over-smoothing. Kaixion Zhou proposes a Group distance ratio to measure distances between different groups with the same label and Instance information gain to measure dependency between node feature and representation. Chen et al. (2019) propose mean-average distance to measure similarity between nodes. More lately Wu et al. (2023) propose a new node similarity measure of the distance between features of nodes and the mean feature as follows

$$\mathcal{E}_W(X) = \|X - 1_{\gamma_X}\|_2^2$$

where $1_{\gamma_X} = \frac{\sum_{i \in [n]} X_i}{n}$. Dirichlet energy is also a commonly used measure as follows

$$\mathcal{E}_D(X) = \sum_{i \in [n]} \sum_{j \in \mathcal{N}_i} \|X_i - X_j\|_2^2$$

Generally Rusch et al. (2023a) defines the following concept of node similarity measure and over-smoothing

**Definition 3.1.** *(Over-smoothing) Let G be an undirected, connected graph, and $X^k \in \mathbb{R}^{n \times m}$ denote the k-th layer hidden features of an N-layer GNN defined on G. Moreover, we call $\mathcal{E} : \mathbb{R}^{n \times m} \to \mathbb{R}_{\geq 0}$ a node similarity measure if it satisfies the following axioms:*

- *$\exists c \in \mathbb{R}^m$ with $X_i = c$ for all nodes $i \in V$ if and only if $\mathcal{E}(X) = 0$, for $X \in \mathbb{R}^{n \times m}$*

- *$\mathcal{E}(X + Y) \leq \mathcal{E}(X) + \mathcal{E}(Y)$, for all $X, Y \in \mathbb{R}^{n \times m}$*

*We then define over-smoothing with respect to $\mu$ as the layer-wise exponential convergence of the node-similarity measure $\mathcal{E}$ to zero, that is for $n = 0, \ldots, N$ and some constants $C_1, C_2 > 0$*

$$\mathcal{E}(X^n) \leq C_1 e^{-C_2 n}$$

If we suppose $1_{\gamma_X} = 0$, then $\mathcal{E}_W(X) = \|X\|_2^2$, this is the energy of zero-order derivative of features. Dirichlet energy can be considered as energy of the first-order derivative of features. Based on these observations, existing measures only capture information on the low-order derivatives, we propose a series of node similarity measures as follows to capture information on higher-order derivatives which is a generalization of the two above measures and proves that attention-based GNNs are over-smoothing under our proposed measure.

**Theorem 3.1.** *$G = (V, E, \omega, \mu)$ is a connected weighted graph, Given a vector-valued function $f$ on the graph, We define the energy of higher-order derivatives of $f$ as follows*

$$\mathcal{E}_m^p(f) := \int_V \|\nabla_\mu^m f\|_p^p d\mu$$

*where $p \geq 1, m \in \mathbb{N}$ and higher-order derivatives of $f$ is defined as follows*

$$\|\nabla_\mu^m {}_p f\| = \begin{cases} \|(-\Delta_\mu)^{\frac{m}{2}} f\|_p & \text{if } m \text{ is an even number} \\ \|\nabla_\mu(-\Delta_\mu)^{\frac{m-1}{2}} f\|_p & \text{if } m \text{ is an odd number} \end{cases}$$

*where $(-\Delta_\mu)^o f = \lim_{s \to 0+} (-\Delta_\mu)^s f = f - \frac{1}{|V|_\mu} \int_V f d\mu$, fractional Laplacian follows the definition in (Maskey et al., 2023). Then node similarities defined as $\gamma_m^p(f) = (\mathcal{E}_m^p(f))^{\frac{1}{p}}$ satisfy two conditions of node-similarity measure. Also, $\mathcal{E}_W$ and $\mathcal{E}_D$ are all special cases of our proposed measures.*

## 3.2 ILLUSTRATION OF PROPOSED MEASURE

We claim that exactly $\mathcal{E}_m^p(f)$ are all equivalent if $m \geq 1$. If additionally assume that the graph is connected, then for all $m \geq 0$, $\mathcal{E}_m^p(f)$ are all equivalent. Case $m \leq 1$ is direct consequence of Poincare inequality and will be left in next section. We give claim for $m \geq 1$ as follows

**Theorem 3.2.** *$G = (V, E, \omega, \mu)$ is a weighted graph. Then for any function f on the graph, there exist constants $C_1, C_2, C_3, C_4 > 0$ such that*

$$C_2 \int \|\nabla_\mu f\|_p^p d\mu \leq \int \|\Delta_\mu f\|_p^p d\mu \leq C_1 \int \|\nabla_\mu f\|_p^p d\mu$$

*and*

$$C_3 \int \|\nabla_\mu \Delta_\mu f\|_p^p d\mu \le \int \|\Delta_\mu f\|_p^p d\mu \le C_4 \int \|\nabla_\mu \Delta_\mu f\|_p^p d\mu$$

*constants $C_2$ will be $\lambda_1$ when $p = 2$. Replace $f$ by $\Delta_\mu^m g$ we can get the equivalence for all $m \ge 1$.*

This theorem implies directly that if GNNs are over-smooth with respect to one of these measures, they will also be over-smooth with respect to other. Equivalence between $\mathcal{E}_m^p$ can be used to estimate the rate of over-smoothing. Given a weighted graph $G = (V, E, \omega, \mu)$, $\lambda_1(-\Delta_\mu)$ is closely related to the trade-off between over-smoothing and over-squashing (Karhadkar et al., 2023). The higher $\lambda_1$, The less over-squashing and the more over-smoothing. Much work is devoted to analyzing relations between $\lambda_1(-\Delta_\mu)$ and over-smoothing (Giraldo et al., 2023), However the existing results are not satisfactory (Chung, 1996)(Cai and Wang, 2020), existing results of decay rate require condition $1 - \lambda_1(-\Delta_\mu) \ge \lambda_N(-\Delta_\mu) - 1$ or self-loop.

Usually, we think that the reason behind message-passing type GNNs is spatial propagation. SGC removes the transformation operation in GCN and achieves competing performance with GCN. Its structure is as follows

$$Y = softmax(\tilde{A}_{sym}^K X W)$$

where $K$ is number of layers and $W$ is learnable parameters. Wang et al. (2021) show that at the stage of feature propagation Simplified graph convolution is equivalent to heat diffusion on the graph with fixed step size $\Delta t = 1$, it is known that features within each connected component propagated by heat diffusion will converge to a constant vector, they control the time step size to address the over-smoothing. However, existing simplified graph convolution-type methods and equation-based methods do not analyze the decay rate. For simplicity and clarity, we only consider the propagation of SGC with general kernel and heat kernel considered by (Wang et al., 2021) here and prove that for a general weighted graph, over-smoothing of them are closely related to $\lambda_1(-\Delta_\mu)$.

**Theorem 3.3.** *Let $G = (V, E, \omega, \mu)$ be a weighted graph. If $f$ is a solution of the heat equation*

$$\frac{\partial f}{\partial t} = \Delta_\mu f$$

*with initial condition $f(t = 0) = f_0$, then there exists constant $C_5 > 0$ such that Dirichlet energy of $f$ will decay exponentially when $t$ tends to infinity*

$$\int_V \|\nabla_\mu f\|_2^2 d\mu \le C_5 e^{-2\lambda_1(-\Delta_\mu)t} \int_V \|\nabla_\mu f_0\|_2^2 d\mu$$

*Suppose further $\forall i \in [n], \sum_{j \in \mathcal{N}_i} \omega_{ij} \le \mu_i$, Let random walk matrix be $P = \Delta_\mu + I$, then for the arbitrary function $f$ on the graph, we have the decay rate as follows*

$$\int_V \|\nabla_\mu P^k f\|_2^2 d\mu \le \left(1 - (2 - \lambda_N(-\Delta_\mu))\lambda_1(-\Delta_\mu)\right)^k \int_V \|\nabla_\mu f\|_2^2 d\mu$$

*where $k \in \mathbb{N}, k \ge 1$. also for $\tilde{A}_{sym}$ we specify $\omega_{ij} = 1, \mu_i = D_i + 1$, we have decay rate*

$$\int_V \|\nabla_\mu(\tilde{D}^{-\frac{1}{2}} \tilde{A}_{sym}^k f)\|_2^2 d\mu \le \left(1 - (2 - \lambda_N(-\tilde{\Delta}_{rw-adj}))\lambda_1(-\tilde{\Delta}_{rw-adj})\right)^k \int_V \|\nabla_\mu(\tilde{D}^{-\frac{1}{2}} f)\|_2^2 d\mu$$

We note that the decay rate on Theorem 3.3 is almost the best we can achieve. If $\lambda_N(-\Delta_\mu) = 2$, then $\int_V \|\nabla_\mu P f\|_2^2 d\mu$ possible equate $\int_V \|\nabla_\mu f\|_2^2 d\mu$. Consider bipartite graph $K_{2,2}$ with nodes $V = \{1, 2, 3, 4\}$ and edges $E = \{(1, 3), (1, 4), (2, 3), (2, 4)\}$, given feature $X$ such that $X(1) = X(2) = 1, X(3) = X(4) = 0$, then $\int_V \|\nabla_\mu A_{rw} X\|_2^2 d\mu = \int_V \|\nabla_\mu X\|_2^2 d\mu$. In conclusion, self-loop is a necessary condition for over-smoothing.

## 4 ADDRESSING OVER-SMOOTHING

In this section, we introduce Poincare inequality on graphs and propose normalization termed Poincarenorm which is a generalization of Pairnorm. Generally, we work on a weighted graph $G = (V, E, \omega, \mu)$ and a vector-valued function $X : V \to \mathbb{R}^d$ is given,

### 4.1 POINCARENORM

First, we recall that PairNorm considers total pairwise square distance(TPSD) as their measure of over-smoothing

$$\text{TPSD}(X) = \sum_{i,j \in [n]} \|X(i) - X(j)\|_2^2$$

TPSD is a special case of Dirichlet energy assuming all nodes are connected. Then they propose PairNorm composed of two steps: centering and scaling

$$X^c(i) = X(i) - \frac{1}{n}\sum_{k=1}^n X(k)$$

$$\tilde{X}(i) = s \frac{X^c(i)}{\sqrt{\frac{1}{n}\sum_{i=1}^n \|X^c(i)\|^2}}$$

This normalization will make $\text{TPSD}(\tilde{X})$ a constant $2n^2s^2$. This consideration has three limitations. First making TPSD a constant will strictly constrain the performance of GNNs. The second limitation is that consideration of TPSD violently assumes all nodes are connected, generally, we have $\mathcal{E}_D(X) \leq \text{TPSD}(X)$ but it is generally not true that $\text{TPSD}(X) \leq C\mathcal{E}_D(X)$ for a constant $C$, so keeping TPSD away from zero can not ensure that $\mathcal{E}_D(X)$ is away from zero. The third limitation is that PairNorm only considers Dirichlet energy type measure, It can not control the energy of higher-order derivatives. To address the first limitation, we first propose a generalization of TPSD. we construct a new weighted graph $\tilde{G} = (V, \tilde{E}, \tilde{\omega}, \tilde{\mu})$ from $G = (V, E, \omega, \mu)$, in $\tilde{G}$ all nodes are connected and $\tilde{\omega}_{ij} = 2$. We propose a total pairwise distance of power p with $p \geq 1$ as follows

$$\text{TPD}_p(X) = \int_V \|\nabla_{\tilde{\mu}}X\|_p^p d\tilde{\mu} = \sum_{i \in [n]} \sum_{j \in [n]} \|X(j) - X(i)\|_p^p$$

TPSD is a special case of $\text{TPD}_p$ when $p = 2$. Similar to previous work we consider normalization on the graph to control $\text{TPD}_p$. To control $\text{TPD}_p$, we only need to guarantee $\text{TPD}_p$ bounded from below and above. We first introduce Poincare inequality on the graph. The existence of Poincare inequality on a graph is closely related to many properties on a graph such as estimation of heat kernel and volume doubling property (Horn et al., 2019). Poincare inequality on a graph is stated as follows

**Theorem 4.1.** $G = (V, E, \omega, \mu)$ *is a weighted finite connected graph. Then there exist constants* $C_6, C_7 > 0$ *such that for any vectored valued function f on a graph, we have the following Poincare inequality with* $p \geq 1$

$$\int_V \|f - \frac{1}{|V|_\mu}\int_V f d\mu\|_p^p d\mu \leq C_6 \int_V \|\nabla_\mu f\|_p^p d\mu$$

*particularly when* $p = 2$*, constant* $\frac{1}{C_6}$ *equates* $\lambda_1(-\Delta_\mu)$*. We also have the following inequalities*

$$\int_V \|\nabla_\mu f\|_p^p d\mu \leq C_7 \int_V \|f - \frac{1}{|V|_\mu}\int_V f d\mu\|_p^p d\mu$$

*thus* $\mathcal{E}_0^p$ *is equivalent to* $\mathcal{E}_1^p$*.*

Poincare inequality is a powerful tool. Notice that if the right-hand side of Poincare inequality is divided by the left-hand side, then Dirichlet-type energy will be bounded from below and above by Theorem 4.1. Based on this observation we define PoincareNorm as follows. Given a vector-valued function $X$, we define the p-PoincareNorm$_0(X)$ of the input $X$ with $p \geq 1$ as follows

$$\text{p-PoincareNorm}_0(X) = s|V|_{\tilde{\mu}}^{\frac{1}{p}} \frac{X - C}{(\int_V \|X - \frac{1}{|V|_{\tilde{\mu}}}\int_V X d\tilde{\mu}\|^p d\tilde{\mu})^{\frac{1}{p}}}$$

where $s$ is the scaling hyperparameter and $C$ is a centering constant which can be set to zero or $\frac{1}{|V|}\int_V X d\tilde{\mu}$. Apparently, this normalization generalizes PairNorm, PairNorm is a special case that $\mu_i = 1$ for all $i \in V$, $p = 2$ and $C = \frac{1}{|V|}\int_V X d\tilde{\mu}$. By the equivalence of $\mathcal{E}_0^p$ and $\mathcal{E}_1^p$, we know that

$$\frac{1}{C_6} \leq \int_V \|\nabla_{\tilde{\mu}}\tilde{X}\|_p^p d\tilde{\mu} \leq C_7$$

To address the second limitation and the third limitation, we first observe that

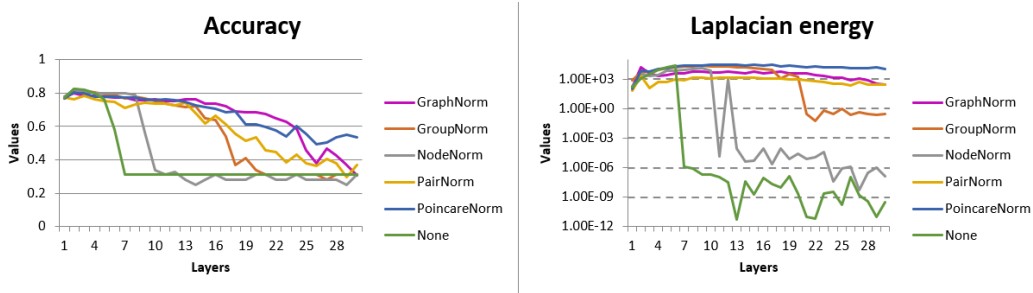

Figure 1: Comparision of different normalization methods enhanced GCN with bias applied on Cora with no missing features

**Theorem 4.2.** *Given a weighted graph $G = (V, E, \omega, \mu)$ and a new weighted graph $\tilde{G} = (V, \tilde{E}, \tilde{\omega}, \tilde{\mu})$ constructed from $G$ where $\tilde{E}$ assume all nodes are connected and $\tilde{\omega}_{ij} = 2$. Then there exist constants $C_8, C_9, C_{10}, C_{11}$ such that we have the following inequalities*

$$TPD_p(\Delta_\mu X) \le C_8 \int_V \|\Delta_\mu X\|_p^p d\mu \le C_9 \big(\mathcal{E}_D(X)\big)^{\frac{p}{2}}$$

*and*

$$\big(\mathcal{E}_D(X)\big)^{\frac{p}{2}} \le C_{10} \int_V \|\Delta_\mu X\|_p^p d\mu \le C_{11} TPD_p(\Delta_\mu X)$$

Therefore to control $\mathcal{E}_D(X)$ and Laplacian energy $\int_V \|\Delta_\mu X\|_p^p d\mu$, we need only to control $TPD_p(\Delta_\mu X)$, similar to previous analysis we define p-PonicareNorm$_1$ as follows

$$\text{p-PonicareNorm}_1(X) = s|V|_{\tilde{\mu}}^{\frac{1}{p}} \frac{X - C}{\left(\int_V \|\Delta_\mu X - \frac{1}{|V|_{\tilde{\mu}}} \int_V \Delta_\mu X d\tilde{\mu}\|^p d\tilde{\mu}\right)^{\frac{1}{p}}}$$

Under this normalization $TPD_p(\Delta_\mu X)$ will have a fixed upper bound and lower bound. Generally, we can define p-PonicareNorm$_m$ of input $X$ with $m \in \mathbb{N}$ as follows

$$\text{p-PonicareNorm}_m(X) = s|V|_{\tilde{\mu}}^{\frac{1}{p}} \frac{X - C}{\left(\int_V \|\Delta_\mu^m X - \frac{1}{|V|_{\tilde{\mu}}} \int_V \Delta_\mu^m X d\tilde{\mu}\|^p d\tilde{\mu}\right)^{\frac{1}{p}}}$$

By previous analysis, we know that p-PonicareNorm$_m$ can control the energy of higher-order derivatives more finely than PairNorm.

## 4.2 ESTIMATION OF POINCARENORM

We term $\mathcal{E}_2^2(X) = \sum_{i \in [n]} \|\Delta_\mu X\|_2^2(i)\mu_i$ with $\omega_{ij} = 1$ and $\mu_i = D_i + 1$ as Laplacian energy. To test the efficiency of p-PonicareNorm$_m$ and its ability to control Laplacian energy, we simply here use GCN with bias as the base model and apply different normalizations on dataset Cora. PoincareNorm is set to be 8-PoincareNorm$_1$. Parameters of other normalization methods are introduced in Appendix A.3 and detailed experiment setups are introduced in Appendix A.5. We experiment in the scenario with no missing features and vary layers from 1 to 30. Each experiment runs 1000 epochs 5 times. We report average performance and the results are plotted in Figure 1, the left figure reports the mean accuracy and the right figure reports the mean Laplacian energy of the output. As the results show, The base model with normalization methods does not perform much better than the base model with no normalization when the neural network is shallow. However, when the neural network goes deep, PoincareNorm outperforms all normalizations and can control Laplacian energy better than others.

## 4.3 TIME COMPLEXITY

Suppose the number of nodes is $n$, the number of edges is $e$, given a feature $X : V \to \mathbb{R}^d$. For p-PoincareNorm$_0$, computational complexity of $\int_V \|X - \frac{1}{|V|} \int_V X d\tilde{\mu}\|_p^p d\tilde{\mu}$ is $\mathcal{O}(nd)$, therefore so is p-PoincareNorm$_0$. For p-PoincareNorm$_m$ when $m \ge 1$, computational complexity of $\int_V \|\Delta_\mu^m X - \frac{1}{|V|} \int_V \Delta_\mu^m X d\tilde{\mu}\|_p^p d\tilde{\mu}$ is $\mathcal{O}(mde)$, therefore so is p-PoincareNorm$_m$

## 5 EXPERIMENTS

In this section, we empirically evaluate the efficiency of our proposed normalization in enabling deep GNN in the scenario with missing features.

### 5.1 EXPERIMENT SET UP

**Datasets**  We conduct our model on three well-known datasets: Cora, Citeseer, and Pubmed(Yang et al., 2016). We use the standard split of the training set and validation set, the remaining nodes are the test set. Details of datasets are introduced in Appendix A.2.

**Model**  Since the graph convolutional network with bias and the graph attention network with bias are more practical compared to generic ones, we use the graph convolutional network with bias, the graph attention network with bias, and simplified graph convolution as the backbone of neural networks. Graph convolutional network with bias uses $\tilde{A}_{rw}$ as the kernel and SGC uses $\tilde{A}_{sym}$ as the kernel. We implement GAT by torch geometric(Fey and Lenssen, 2019). Details of the base model are introduced in Appendix A.4.

**Baselines**  We compare our method with no normalization, PairNorm, GroupNorm, NodeNorm, and GraphNorm.

**Hyperparameter**  We set hidden features as 64 for GCN with bias and GAT with bias. Hyperparameters of baselines are specified in Appendix A.3. We use Adam optimizer (Kingma and Ba, 2017), the learning rate is 0.01, L2 regularization is 5e-4, and the dropout rate is 0.6. In the scenario with the missing features, we apply GCN with bias and GAT with bias with varying layers from $\{1, 2, \ldots, 20, 25, 30\}$, SGC with varying layers from $\{1, 2, \ldots, 10, 15, 20, \ldots, 50\}$. the main hyperparameter in PoincareNorm is order of derivative $m$, power $p$, constant $C$, scale $s$, edge weight $\omega_{ij}$ and node weights $\mu_i$. We vary $m$ from $\{0, 1\}$ and $p$ from $\{4, 8\}$. For all datasets we set all $\omega_{ij} = 1$, $C = \frac{1}{|V|_{\tilde{\mu}}} \int_V X d\tilde{\mu}$, $s = 1$ for input $X$. For Cora and Pubmed we set $\tilde{\mu}_i = \mu_i = \sum_{j \in \mathcal{N}_i} \omega_{ij}$, for Citeseer we set $\tilde{\mu}_i = \mu_i = 1 + \sum_{j \in \mathcal{N}_i} \omega_{ij}$.

**Configurations**  We apply each normalization method after the graph convolutional layer and before the nonlinear activation layer. For each normalization method, we run the experiment with 1000 epochs 5 times for each layer and report the average accuracy and standard variation of the layer which achieve the best mean validation accuracy.

Table 1: Comparison of different normalization methods enhanced GCN with bias applied on datasets

| Dataset
Method | Cora
Acc | Citeseer
Acc | Pubmed
Acc |
|---|---|---|---|
| None | 0.6512(0.0074) | 0.3576(0.0079) | 0.5396(0.0460) |
| PairNorm | 0.7463(0.0379) | 0.4330(0.0233) | 0.7063(0.0054) |
| GroupNorm | 0.6641(0.0326) | 0.3855(0.0419) | 0.5814(0.0134) |
| GraphNorm | 0.7594(0.0169) | 0.4587(0.0169) | 0.6949(0.0238) |
| NodeNorm | 0.7385(0.0060) | 0.4487(0.0147) | 0.5930(0.0080) |
| 4-PoincareNorm$_0$ | 0.7427(0.0273) | 0.4389(0.0296) | 0.6799(0.0213) |
| 8-PoincareNorm$_0$ | 0.7603(0.0083) | 0.4328(0.0257) | 0.6875(0.0222) |
| 4-PoincareNorm$_1$ | 0.7631(0.0073) | **0.4602(0.0077)** | **0.7077(0.0156)** |
| 8-PoincareNorm$_1$ | **0.7674(0.0060)** | 0.4572(0.0174) | 0.6898(0.0094) |

### 5.2 EXPERIMENTS RESULT

Scenario with missing features in common in the real world such as missing features of users on social recommendation networks. This scenario is complex and require more layers to learn information of larger neighborhood compared to the classical scenario with no missing features. More specifically,

Let $M$ be a subset of a set of nodes $V_1$, we remove features of nodes in $M$ and set features as zero. The fraction $\frac{|M|}{|V_1|}$ is called the missing rate, where $|M| = \sum_{i \in M} 1$. To study our proposed normalization to enable neural networks to go deeper, we remove all features in the validation set and test set of datasets and set them to zeros while keeping features in the training set original. Results are reported in Table 1, Table 2, and Table 3 respectively. OOM represents that the experiment is out of memory when GNNs are going deep. As the results show, our proposed normalization outperforms in 6 experiments out of 9 experiments.

Table 2: Comparison of different normalization methods enhanced SGC applied on datasets

| Dataset | Cora | Citeseer | Pubmed |
| Method | Acc | Acc | Acc |
| --- | --- | --- | --- |
| None | 0.6701(0.0068) | 0.3554(0.0232) | 0.6581(0.0110) |
| PairNorm | 0.7691(0.0012) | 0.4656(0.0007) | 0.6923(0.0054) |
| GroupNorm | OOM | OOM | OOM |
| GraphNorm | 0.7620(0.0063) | OOM | OOM |
| NodeNorm | 0.7574(0.0159) | **0.5317(0.0086)** | **0.7773(0.0023)** |
| 4-PoincareNorm$_0$ | 0.7854(0.0012) | 0.4926(0.0261) | 0.7017(0.0150) |
| 8-PoincareNorm$_0$ | 0.7846(0.0015) | 0.4834(0.0178) | 0.7001(0.0048) |
| 4-PoincareNorm$_1$ | 0.7971(0.0035) | 0.5039(0.0237) | 0.7012(0.0125) |
| 8-PoincareNorm$_1$ | **0.7975(0.0016)** | 0.5229(0.0181) | 0.6998(0.0041) |

Table 3: Comparison of different normalization methods enhanced GAT with bias applied on datasets

| Dataset | Cora | Citeseer | Pubmed |
| Method | Acc | Acc | Acc |
| --- | --- | --- | --- |
| None | 0.7063(0.0086) | 0.4096(0.0164) | 0.5386(0.0353) |
| PairNorm | 0.7073(0.0504) | 0.4351(0.0278) | 0.6469(0.0218) |
| GroupNorm | 0.7242(0.0141) | 0.4208(0.0363) | 0.6183(0.0033) |
| GraphNorm | 0.7351(0.0017) | 0.4083(0.0149) | 0.6183(0.0176) |
| NodeNorm | 0.7477(0.0038) | **0.4781(0.0176)** | 0.6240(0.0057) |
| 4-PoincareNorm$_0$ | 0.7297(0.0159) | 0.3932( 0.0234) | **0.6504(0.0338)** |
| 8-PoincareNorm$_0$ | 0.7095(0.0220) | 0.4027(0.0300) | 0.6477(0.0279) |
| 4-PoincareNorm$_1$ | **0.7507(0.0053)** | 0.4185(0.0154) | 0.6440(0.0195) |
| 8-PoincareNorm$_1$ | 0.7443(0.0045) | 0.4323(0.0104) | 0.6460(0.0322) |

# 6 CONCLUSION

In this work, we generalize some existing node similarity measures including Dirichlet energy, and propose measures called the energy of higher-order derivatives. We rigorously establish relations between the energy of higher-order derivatives. Using this relation we establish the decay rate of Dirichlet energy of diffusion under heat kernel and discrete random walk, and we show that self-loop is a necessary condition for over-smoothing. These results and techniques can also be easily used to establish the decay rate for other dynamical systems. Future work can design GNNs in light of these measures. To address over-smoothing in light of the energy of higher-order derivatives, we propose a normalization termed PoincareNorm which is a generalization of PairNorm. PoincareNorm outperforms existing normalizations on the semi-supervised node classification task in the scenario with missing features and can control the energy of higher-order derivatives well.

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

# A  APPENDIX

# B  EXPERIMENT DETAILS

## B.1  RUNNING ENVIRONMENT

All our normalization methods and base models are implemented in PyTorch. Experiment that apply GCN with bias with different normalization methods on dataset Cora with no missing features and experiment that apply GAT with bias with different normalization methods on dataset Citeseer with 100% missing features are conducted on a machine with NVIDIA GeForce RTX 3070 Ti Laptop 16

GB GPU and 12th Gen Intel(R) Core(TM) i7-12700H CPU. Other experiments are conducted on a machine with NVIDIA GeForce RTX 3080 Laptop 16 GB GPU and AMD Ryzen 9 5900HX with Radeon Graphics CPU.

## B.2 DATASET STATISTICS

Table 4: Dataset Statistics

| Datasets | Cora | Citeseer | Pubmed |
|---|---|---|---|
| Nodes | 2708 | 3327 | 19717 |
| Edges | 5429 | 4732 | 44338 |
| Features | 1433 | 3703 | 500 |
| Classes | 7 | 6 | 3 |
| Training Nodes | 140 | 120 | 60 |
| Validation Nodes | 500 | 500 | 500 |
| Test Nodes | 2068 | 2707 | 19157 |
| Label Rate | 0.052 | 0.036 | 0.003 |

## B.3 DETAILS OF BASELINE

We use the implementation released by authors to implement baseline

- GroupNorm:https://github.com/Kaixiong-Zhou/DGN
- PairNorm:https://github.com/LingxiaoShawn/PairNorm
- GraphNorm:https://github.com/lsj2408/GraphNorm
- NodeNorm:https://github.com/miafei/NodeNorm

We give details of each baseline and hyperparameter choice as follows

**GroupNorm**   Given an input $H^k \in \mathbb{R}^{n \times d}$, then GroupNorm comprise of two steps. First differentiable cluster nodes into $l$ groups

$$S^k = softmax(H^k U^k)$$

where $U^k \in \mathbb{R}^{d \times l}$. Then generate $l$ features

$$H_i^k = S^k[:, i] \circ H^k \ \ i = 1, \dots, l$$

and normalization with each group

$$\tilde{H}_i^k = \gamma_i(\frac{H_i^k - \mu_i}{\sigma_i}) + \beta_i$$

where $\gamma_i$ and $\beta_i$ are learnable parameters. $\mu_i$ and $\sigma_i$ are mean and variation over nodes within the same dimension of features. Finally, GroupNorm generates the final embedding

$$\tilde{H}^k = H^k + \lambda \sum_{i=1}^{l} \tilde{H}_i^k$$

For all experiments in this paper. The number of groups $l$ in GroupNorm is 5 for Pubmed and 10 for others, skip weight $\lambda$ for GroupNorm is set to be 0.003.

**NodeNorm**   Given an input $H^k \in \mathbb{R}^{n \times d}$, its i-th row and j-th column item is denoted as $H_{i,j}^k$. NodeNorm can be expressed as

$$NodeNorm(H_i^k) = \frac{H_i^k}{(\sigma_i)^p}$$

where $\mu_i = \frac{\sum_{f=1}^{d} H_{i,j}^k}{d}$ and $\sigma_i^2 = \frac{\sum_{j=1}^{n} (H_{i,j}^k - \mu_i)^2}{d}$. For all experiments, we set $p = 2$.

**PairNorm**  Given an input $H \in \mathbb{R}^{n \times d}$, its i-th row vector is denoted as $H_i^k$. Then PairNorm can be expressed as

$$H_i^c = H_i - \frac{1}{n} \sum_{k=1}^n H_k$$

$$\tilde{H}_i = s \frac{H_i^c}{\sqrt{\frac{1}{n} \sum_{i=1}^n \|H_i^c\|^2}}$$

For all experiments, we set $s = 1$.

**GraphNorm**  Given an input $H^k \in \mathbb{R}^{n \times d}$, its i-th row and j-th column item is denoted as $H_{i,j}^k$. Then GraphNorm can be expressed as

$$GraphNorm(H_{i,j}^k) = \gamma_j \frac{H_{i,j}^k - \alpha_j \mu_j}{\sigma_j} + \beta_j$$

where $\gamma_j, \beta_j, \alpha_j$ are learnable parameters. $\mu_j = \frac{\sum_{i=1}^n H_{i,j}^k}{n}$, $\sigma_j^2 = \frac{\sum_{i=1}^n (H_{i,j}^k - \alpha_j \mu_j)^2}{n}$

### B.4 Details of base model

Table 5: Graph convolutional layer of base models

| Models | aggregation function |
|---|---|
| GAT with bias | $X_i^{k+1} = \sum_{j \in \mathcal{N}_i \cup i} a_{ij} W^k X_j^k + b^k$ |
| GCN with bias | $X^{k+1} = \tilde{A}_{rw} X^k W^k + b^k$ |
| SGC | $X^{k+1} = \tilde{A}_{sym} X^k$ |

### B.5 Experiment set up for estimation of PoincareNorm

For parameters of PoincareNorm, we set $m = 1, p = 8, \omega_{ij} = 1, \tilde{\mu}_i = \mu_i = \sum_{j \in \mathcal{N}_i} \omega_{ij}, C = \frac{1}{|V|_{\tilde{\mu}}} \int_V X d\tilde{\mu}$ and $s = 1$. We use Adam optimizer(Kingma and Ba, 2017), the learning rate is 0.01, L2 regularization is 5e-4, and the dropout rate is 0.6. All normalization methods are after the graph convolutional layer and before the nonlinear activation layer.

## C Proofs of theorems

### C.1 Proof of Theorem 2.1

Given a weighted graph $G = (V, E, \omega, \mu)$ and vectored-valued functions $f, g : V \to \mathbb{R}^d$. First, we suppose $d = 1$, we have the following equality

$$\sum_{i \in [n]} \sum_{j \in \mathcal{N}_i} \omega_{ij} f(i) g(i)$$

$$= \sum_{i \sim j} \omega_{ij} \big(f(i)g(i) + f(j)g(j)\big)$$

$$= \sum_{i \in [n]} \sum_{j \in \mathcal{N}_i} \omega_{ij} f(j) g(j)$$

and

$$\sum_{i \in [n]} \sum_{j \in \mathcal{N}_i} \omega_{ij} f(j) g(i)$$

$$= \sum_{i \sim j} \omega_{ij} \big(f(j)g(i) + f(i)g(j)\big)$$

$$= \sum_{i \in [n]} \sum_{j \in \mathcal{N}_i} \omega_{ij} f(i) g(j)$$

Combining above equalities we deduce

$$
\begin{aligned}
\int_V \Delta_\mu f \cdot g d\mu &= \sum_{i=1}^n \sum_{j \in \mathcal{N}_i} \omega_{ij}\big(f(j) - f(i)\big)g(i) \\
&= \sum_{i=1}^n \sum_{j \in \mathcal{N}_i} \big(\omega_{ij} f(j)g(i) - \omega_{ij} f(i)g(i)\big) \\
&= \sum_{i=1}^n \sum_{j \in \mathcal{N}_i} \big(\omega_{ij} f(i)g(j) - \omega_{ij} f(j)g(j)\big) \\
&= \sum_{i=1}^n \sum_{j \in \mathcal{N}_i} \omega_{ij}\big(f(i) - f(j)\big)g(j)
\end{aligned}
$$

Thus adding the right-hand side of the first line and last line we get

$$
\begin{aligned}
\int_V \Delta_\mu f \cdot g d\mu &= \sum_{i=1}^n \sum_{j \in \mathcal{N}_i} \frac{\omega_{ij}}{2}(f(j) - f(i))(g(i) - g(j)) \\
&= -\int_V \nabla_\mu f \cdot \nabla_\mu g
\end{aligned}
$$

Now suppose general $d$ and $f = (f_1, \ldots, f_d)^T, g = (g_1, \ldots, g_d)^T$, then a direct calculation yield

$$
\begin{aligned}
&\int_V \Delta_\mu f \cdot g d\mu \\
&= \int_V \Delta_\mu f_1 \cdot g_1 d\mu + \cdots + \Delta_\mu f_d \cdot g_d d\mu \\
&= -\int_V \nabla_\mu f_1 \cdot \nabla_\mu g_1 d\mu - \cdots - \nabla_\mu f_d \cdot \nabla_\mu g_d d\mu \\
&= -\int_V \nabla_\mu f \cdot \nabla_\mu g d\mu
\end{aligned}
$$

## C.2 Proof of Theorem 2.2

Suppose $0 = \tilde{\lambda}_0 \leq \cdots \leq \tilde{\lambda}_{n-1}$ are eigenvalues and $v_0, \ldots, v_{n-1}$ are eigenvectors corresponding to eigenvalues respectively such that $\int_V \|v_k\|_2^2 d\mu = 1, \ \forall 1 \leq k \leq n$ and $\int_V v_i \cdot v_j d\mu = 0, \ \forall i \neq j$. For any vector-valued function $f : V \to \mathbb{R}^d$, suppose $f = c_0 v_0 + \cdots + c_{n-1} v_{n-1}$, then we have

$$
\begin{aligned}
\int_V \|\nabla_\mu f\|_2^2 d\mu &= \int_V -\Delta_\mu f \cdot f d\mu \\
&= \tilde{\lambda}_0 c_0^2 v_0 + \cdots + \tilde{\lambda}_{n-1} c_{n-1}^2 v_{n-1} \\
&\leq \lambda_N (c_0^2 v_0 + \cdots + c_{n-1}^2 v_{n-1}) \\
&= \lambda_N \int_V \|f\|_2^2
\end{aligned}
$$

Obviously we have $\int_V \|\nabla_\mu v_{n-1}\|_2^2 d\mu = \lambda_N \int_V \|v_{n-1}\|_2^2$. We compute that for all vector-valued function $g : V \to \mathbb{R}^d$

$$
\begin{aligned}
\int_V \|\nabla_\mu g\|_2^2 d\mu &= \sum_{i \in [n]} \Big( \sum_{j \in \mathcal{N}_i} \frac{1}{2} \omega_{ij} \|g(j) - g(i)\|_2^2 \Big) \\
&\leq \sum_{i \in [n]} \Big( \sum_{j \in \mathcal{N}_i} \omega_{ij} (\|g(j)\|_2^2 + \|g(i)\|_2^2) \Big) \\
&= 2 \sum_{i \in [n]} \Big( \frac{\sum_{j \in \mathcal{N}_i} \omega_{ij}}{\mu_i} \Big) \|g(i)\|_2^2 \mu_i \\
&\leq 2 M_{max} \int_V \|g\|_2^2 d\mu
\end{aligned}
$$

Therefore we have $\lambda_N \leq 2M_{max}$

### C.3 Proof of Theorem 3.1

(1) Suppose $X : V \to \mathbb{R}^d$ is a constant vector-valued function on the connected graph $G = (V, E, \omega, \mu)$, we wish to prove that $\mathcal{E}_m^p(X) = 0$. Case $m \leq 1$ is well known. For Case $m \geq 2$, we compute that

$$
\Delta_\mu X(i) = \frac{\omega_{ij} \big( X(j) - X(i) \big)}{\mu_i} = 0
$$

Therefore if $m$ is even, $\mathcal{E}_m^p(X) = \int_V \|\Delta_\mu^{\frac{m}{2}} X\|_p^p d\mu = 0$. If $m$ is odd, $\mathcal{E}_m^p(X) = \int_V \|\nabla_\mu \Delta_\mu^{\frac{m-1}{2}} X\|_p^p d\mu = 0$. Now Suppose that for some $m, p$, $\mathcal{E}_m^p(X) = 0$. we prove inductively that $X$ is a constant vector on the graph. First, suppose $d = 1$, case $m = 1$ is obvious. If $m = 2$, then $\Delta_\mu X = 0$. Suppose $X$ attains its maximum at node i, then we have

$$
0 = \Delta_\mu X(i) = \sum_{j \in \mathcal{N}_i} \frac{\omega_{ij}}{\mu_I} \big( X(j) - X(i) \big) \leq 0
$$

From this we know that for all $j \in \mathcal{N}_i$, $X(j) = X(i)$. Because $G$ is connected, we conclude that for all $k \in V$, $X_k = X_i$. thus $X$ is a constant. If $m = 3$, then $\Delta_\mu X$ is a constant, suppose $\Delta_\mu X = c$. If $c > 0$, then similarly we suppose X attains its maximum at node i, then

$$
c = \Delta_\mu X(i) = \sum_{j \in \mathcal{N}_i} \frac{\omega_{ij}}{\mu_I} \big( X(j) - X(i) \big) \leq 0
$$

this is a contradiction. If $c < 0$, then similarly we suppose X attains its minimum at node i, then

$$
c = \Delta_\mu X(i) = \sum_{j \in \mathcal{N}_i} \frac{\omega_{ij}}{\mu_I} \big( X(j) - X(i) \big) \leq 0
$$

this is also a contradiction. Thus $c = 0$, from the analysis when $m = 2$ we know that $X$ is a constant. For general $m$, if $m$ is even, then $\Delta_\mu^{\frac{m}{2}} X = 0$, using results when $m = 2, 3$ inductively we conclude $X$ is a constant vector. If $m$ is odd, then $\Delta_\mu^{\frac{m-1}{2}} X$ is a constant vector, using results when $m = 2, 3$, we conclude that $X$ is a constant.

Now suppose for general dimension $d$, $X = (X_1, \ldots, X_d)^T$. Then for arbitrary $m$, if $m$ is even. $\Delta_\mu^{\frac{m}{2}} X = (\Delta_\mu^{\frac{m}{2}} X_1, \ldots, \Delta_\mu^{\frac{m}{2}} X_d)^T = 0$. Therefore for all $k$, $\Delta_\mu^{\frac{m}{2}} X_k = 0$, thus $X_k$ is a constant, so $X$ is a constant vector. If $m$ is odd, the proof is similar.

(2) For arbitrary $m, p$ we wish to prove

$$
\gamma_m^p(X^1 + X^2) \leq \gamma_m^p(X^1) + \gamma_m^p(X^2)
$$

For $m = 0$, we compute by Minkowski inequality

$$\gamma_0^p(X^1 + X^2) = \Big(\int_V \|(X^1 + X^2) - \frac{1}{|V|_\mu}\int_V (X^1 + X^2)d\mu\|_p^p d\mu\Big)^{\frac{1}{p}}$$

$$= \Big(\sum_{i\in[n]}\sum_{1\le k\le d}|X_k^1 - \frac{1}{|V|_\mu}\int_V X_k^1 d\mu + X_k^2 - \frac{1}{|V|_\mu}\int_V X_k^2 d\mu|^p\mu_i\Big)^{\frac{1}{p}}$$

$$\le \Big(\sum_{i\in[n]}\sum_{1\le k\le d}(|X_k^1 - \frac{1}{|V|_\mu}\int_V X_k^1 d\mu| + |X_k^2 - \frac{1}{|V|_\mu}\int_V X_k^2 d\mu|)^p\mu_i\Big)^{\frac{1}{p}}$$

$$\le \Big(\sum_{i\in[n]}\sum_{1\le k\le d}(|X_k^1 - \frac{1}{|V|_\mu}\int_V X_k^1 d\mu|)^p\mu_i\Big)^{\frac{1}{p}} + \Big(\sum_{i\in[n]}\sum_{1\le k\le d}(|X_k^2 - \frac{1}{|V|_\mu}\int_V X_k^2 d\mu|)^p\mu_i\Big)^{\frac{1}{p}}$$

$$= \gamma_0^p(X^1) + \gamma_0^p(X^2)$$

For $m = 1$, we compute by Minkowski inequality

$$\gamma_1^p(X^1 + X^2) = \Big(\int_V \|\nabla_\mu(X^1 + X^2)\|_p^p d\mu\Big)^{\frac{1}{p}}$$

$$= \Big(\sum_{i\in[n]}\sum_{j\in\mathcal{N}_i}\sum_{1\le k\le d}\frac{\omega_{ij}}{2}|X_k^1(j) - X_k^1(i) + X_k^2(j) - X_k^2(i)|^p\mu_i\Big)^{\frac{1}{p}}$$

$$\le \Big(\sum_{i\in[n]}\sum_{j\in\mathcal{N}_i}\sum_{1\le k\le d}((\frac{\omega_{ij}}{2})^{\frac{1}{p}}|X_k^1(j) - X_k^1(i)| + (\frac{\omega_{ij}}{2})^{\frac{1}{p}}|X_k^2(j) - X_k^2(i)|)^p\mu_i\Big)^{\frac{1}{p}}$$

$$\le \Big(\sum_{i\in[n]}\sum_{j\in\mathcal{N}_i}\sum_{1\le k\le d}((\frac{\omega_{ij}}{2})^{\frac{1}{p}}|X_k^1(j) - X_k^1(i)|)^p\mu_i\Big)^{\frac{1}{p}} + \Big(\sum_{i\in[n]}\sum_{j\in\mathcal{N}_i}\sum_{1\le k\le d}((\frac{\omega_{ij}}{2})^{\frac{1}{p}}|X_k^2(j) - X_k^2(i)|)^p\mu_i\Big)^{\frac{1}{p}}$$

$$= \gamma_1^p(X^1) + \gamma_1^p(X^2)$$

For $m = 2$, we compute by Minkowski inequality

$$\gamma_2^p(X^1 + X^2) = \Big(\int_V \|\Delta_\mu(X^1 + X^2)\|_p^p d\mu\Big)^{\frac{1}{p}}$$

$$= \Big(\sum_{i\in[n]}\|\Delta_\mu X^1(i) + \Delta_\mu X^2(i)\|_p^p\mu_i\Big)^{\frac{1}{p}}$$

$$= \Big(\sum_{i\in[n]}\sum_{1\le k\le d}|\Delta_\mu X_k^1(i) + \Delta_\mu X_k^2(i)|_p^p\mu_i\Big)^{\frac{1}{p}}$$

$$= \Big(\sum_{i\in[n]}\sum_{1\le k\le d}|\mu_i^{\frac{1}{p}}\Delta_\mu X_k^1(i) + \mu_i^{\frac{1}{p}}\Delta_\mu X_k^2(i)|^p\Big)^{\frac{1}{p}}$$

$$\le \Big(\sum_{i\in[n]}\sum_{1\le k\le d}|\mu_i^{\frac{1}{p}}\Delta_\mu X_k^1(i)|^p\Big)^{\frac{1}{p}} + \Big(\sum_{i\in[n]}\sum_{1\le k\le d}|\mu_i^{\frac{1}{p}}\Delta_\mu X_k^2(i)|^p\Big)^{\frac{1}{p}}$$

$$= \gamma_2^p(X^1) + \gamma_2^p(X^2)$$

For general $m \ge 3$, if $m$ is an even number, then

$$\gamma_m^p(X^1 + X^2) = \Big(\int_V \|\Delta_\mu^{\frac{m}{2}}(X^1 + X^2)\|_p^p d\mu\Big)^{\frac{1}{p}}$$

$$= \Big(\int_V \|\Delta_\mu(\Delta_\mu^{\frac{m}{2}-1}X^1 + \Delta_\mu^{\frac{m}{2}-1}X^2)\|_p^p d\mu\Big)^{\frac{1}{p}}$$

$$\le \Big(\int_V \|\Delta_\mu(\Delta_\mu^{\frac{m}{2}-1}X^1)\|_p^p d\mu\Big)^{\frac{1}{p}} + \Big(\int_V \|\Delta_\mu(\Delta_\mu^{\frac{m}{2}-1}X^2)\|_p^p d\mu\Big)^{\frac{1}{p}}$$

$$= \gamma_m^p(X^1) + \gamma_m^p(X^2)$$

If $m$ is an odd number, then

$$\gamma_m^p(X^1 + X^2) = \Big(\int_V \|\nabla_\mu \Delta_\mu^{\frac{m-1}{2}}(X^1 + X^2)\|_p^p d\mu\Big)^{\frac{1}{p}}$$

$$= \Big(\int_V \|\nabla_\mu(\Delta_\mu^{\frac{m-1}{2}} X^1 + \Delta_\mu^{\frac{m-1}{2}} X^2)\|_p^p d\mu\Big)^{\frac{1}{p}}$$

$$\leq \Big(\int_V \|\nabla_\mu(\Delta_\mu^{\frac{m-1}{2}} X^1)\|_p^p d\mu\Big)^{\frac{1}{p}} + \Big(\int_V \|\nabla_\mu(\Delta_\mu^{\frac{m-1}{2}} X^2)\|_p^p d\mu\Big)^{\frac{1}{p}}$$

$$= \gamma_m^p(X^1) + \gamma_m^p(X^2)$$

## C.4 Proof of Theorem 3.2

We assume $G = (V, E, \omega, \mu)$ is connected first and prove following lemma

**Lemma C.1.** *Given a weighted graph $G = (V, E, \omega, \mu)$ and any vector-valued function $g : V \to \mathbb{R}^d$, there exits constants $C_{12}, C_{13}, C_{14}, C_{15}$ such that the following inequalities hold*

$$C_{12}\Big(\int_V \|g\|_p^p d\mu\Big)^{\frac{1}{p}} \leq \Big(\int_V \|g\|_2^2 d\mu\Big)^{\frac{1}{2}} \leq C_{13}\Big(\int_V \|g\|_p^p d\mu\Big)^{\frac{1}{p}} \tag{1}$$

$$C_{14}\Big(\int_V \|\nabla_\mu g\|_p^p d\mu\Big)^{\frac{1}{p}} \leq \Big(\int_V \|\nabla_\mu g\|_2^2 d\mu\Big)^{\frac{1}{2}} \leq C_{15}\Big(\int_V \|\nabla_\mu g\|_p^p d\mu\Big)^{\frac{1}{p}} \tag{2}$$

*Proof.* Suppose $g = (g_1, \ldots, g_d)^T$. For inequalities (1), if $p > 2$, for the right-hand side of inequality, we compute that

$$\int_V \|g\|_2^2 d\mu = \sum_{i \in [n]} \Big(\sum_{k=1}^d g_k^2(i)\Big)\mu_i$$

$$\leq \sum_{i \in [n]} \Big(\sum_{k=1}^d |g_k(i)|^p\Big)^{\frac{2}{p}} d^{\frac{p-2}{p}} \mu_i$$

$$\leq (dn)^{\frac{p-2}{p}} \Big(\sum_{i \in [n]} \Big(\sum_{k=1}^d |g_k(i)|^p\Big)\mu_i^{\frac{p}{2}}\Big)^{\frac{2}{p}}$$

$$\leq (nd)^{\frac{p-2}{p}} (\mu_{max})^{\frac{p}{2}-1} \Big(\sum_{i \in [n]} \Big(\sum_{k=1}^d |g_k(i)|^p\Big)\mu_i\Big)^{\frac{2}{p}}$$

$$= (nd)^{\frac{p-2}{p}} (\mu_{max})^{\frac{p}{2}-1} \Big(\int_V \|g\|_p^p d\mu\Big)^{\frac{2}{p}}$$

The second line and the third line are from Holder inequality. For the left-hand side of inequality, we compute that

$$\int_V \|g\|_p^p d\mu = \sum_{i \in [n]} \Big(\sum_{k=1}^d |g_k(i)|^p\Big)\mu_i$$

$$\leq \sum_{i \in [n]} \Big(\sum_{k=1}^d |g_k(i)|^2\Big)\mu_i \times \max_{j,i}\big(|g_j(i)|^{p-2}\big)$$

$$= \int_V \|g\|_2^2 d\mu \times \max_{j,i}\big(|g_j(i)|^{p-2}\big)$$

$$\leq \frac{1}{(\mu_{max})^{\frac{p-2}{2}}} \int_V \|g\|_2^2 d\mu \times \Big(\int_V \|g\|_2^2 d\mu\Big)^{\frac{p-2}{2}}$$

$$= \frac{1}{(\mu_{max})^{\frac{p-2}{2}}} \Big(\int_V \|g\|_2^2 d\mu\Big)^{\frac{p}{2}}$$

Proof for case $p < 2$ is similar. For inequalities (2), if $p > 2$, for the right-hand side , suppose the maximal degree is $D_{max} = \max_i D_i$, then we directly compute that

$$
\begin{aligned}
\int_V \|\nabla_\mu g\|_2^2 d\mu &= \sum_{i \in [n]} \sum_{j \in \mathcal{N}_i} \frac{1}{2} \omega_{ij} \|g(j) - g(i)\|_2^2 \\
&= \sum_{i \in [n]} \sum_{j \in \mathcal{N}_i} \frac{1}{2} \omega_{ij} \Big( \sum_{1 \le k \le d} \big( g_k(j) - g_k(i) \big)^2 \Big) \\
&\le \sum_{i \in [n]} \sum_{j \in \mathcal{N}_i} \frac{1}{2} \omega_{ij} \Big( \sum_{1 \le k \le d} |g_k(j) - g_k(i)|^p \Big)^{\frac{2}{p}} d^{\frac{p-2}{p}} \\
&\le \frac{1}{2} \sum_{i \in [n]} \Big( \sum_{j \in \mathcal{N}_i} \omega_{ij}^{\frac{p}{2}} \big( \sum_{1 \le k \le d} |g_k(j) - g_k(i)|^p \big) \Big)^{\frac{2}{p}} (dD_{max})^{\frac{p-2}{p}} \\
&\le \frac{1}{2} \Big( \sum_{i \in [n]} \sum_{j \in \mathcal{N}_i} \omega_{ij}^{\frac{p}{2}} \big( \sum_{1 \le k \le d} |g_k(j) - g_k(i)|^p \big) \Big)^{\frac{2}{p}} (dnD_{max})^{\frac{p-2}{p}} \\
&\le 2^{\frac{2}{p} - 1} \Big( \sum_{i \in [n]} \sum_{j \in \mathcal{N}_i} \frac{1}{2} \omega_{ij} \big( \sum_{1 \le k \le d} |g_k(j) - g_k(i)|^p \big) \Big)^{\frac{2}{p}} (dnD_{max}\omega_{max})^{\frac{p-2}{p}} \\
&= 2^{\frac{2}{p} - 1} (dnD_{max}\omega_{max})^{\frac{p-2}{p}} \Big( \int_V \|\nabla_\mu g\|_p^p d\mu \Big)^{\frac{2}{p}}
\end{aligned}
$$

The third line and the fifth line are from Holder inequality. For the left-hand side we directly compute that

$$
\begin{aligned}
\int_V \|\nabla_\mu g\|_p^p d\mu &= \sum_{i \in [n]} \sum_{j \in \mathcal{N}_i} \frac{1}{2} \omega_{ij} \|g(j) - g(i)\|_p^p \\
&= \sum_{i \in [n]} \sum_{j \in \mathcal{N}_i} \frac{1}{2} \omega_{ij} \Big( \sum_{1 \le k \le d} |g_k(j) - g_k(i)|^p \Big) \\
&\le \sum_{i \in [n]} \sum_{j \in \mathcal{N}_i} \frac{1}{2} \omega_{ij} \Big( \sum_{1 \le k \le d} |g_k(j) - g_k(i)|^2 \Big) \times \max_{i \in [n], j \in \mathcal{N}_i} |g_k(j) - g_k(i)|^{p-2} \\
&\le \int_V \|\nabla_\mu g\|_2^2 d\mu \times \Big( \int_V \|\nabla_\mu g\|_2^2 d\mu \Big)^{\frac{p-2}{2}} \Big( \frac{1}{2} \omega_{max} \Big)^{-\frac{p-2}{2}} \\
&= \Big( \frac{1}{2} \omega_{max} \Big)^{-\frac{p-2}{2}} \Big( \int_V \|\nabla_\mu g\|_2^2 d\mu \Big)^{\frac{p}{2}}
\end{aligned}
$$

case $1 \le p < 2$ is similar. $\qquad\square$

We continue the proof of Theorem 3.2. The proof requires Theorem 4.1 so we assume Theorem 4.1 is correct first for clarity. One can also see the proof of Theorem 4.1 first. Given a weighted graph $G = (V, E, \omega, \mu)$ and a vector-valued function $f : V \to \mathbb{R}^d$, First we wish to prove

$$
C_2 \int \|\nabla_\mu f\|_p^p d\mu \le \int \|\Delta_\mu f\|_p^p d\mu \le C_1 \int \|\nabla_\mu f\|_p^p d\mu \tag{3}
$$

By lemma B.1 we only need to prove for the case $p = 2$. First, we suppose $d = 1$, for the right-hand side of inequality we directly compute that

$$\int_V \|\Delta_\mu f\|_2^2 d\mu = \sum_{i \in [n]} \Big(\frac{\omega_{ij}\big(f(j) - f(i)\big)}{\mu_i}\Big)^2 \mu_i$$

$$= \sum_{i \in [n]} \Big(\frac{\omega_{ij}\big(f(j) - f(i)\big)}{\mu_i^1}\Big)^2 \frac{(\mu_i^1)^2}{\mu_i}$$

$$\leq \sum_{i \in [n]} \omega_{ij}\big(f(j) - f(i)\big)^2 \frac{\mu_i^1}{\mu_i}$$

$$\leq 2 M_{max} \int_V \|\nabla_\mu f\|_2^2 d\mu$$

The third line is from Jenson inequality. For the left-hand side, we have

$$\int_V \|\nabla_\mu f\|_2^2 d\mu = \int_V \|\nabla_\mu (f - \frac{1}{|V|_\mu} \int_V f d\mu)\|_2^2 d\mu$$

$$= -\int_V \Delta_\mu (f - \frac{1}{|V|_\mu} \int_V f d\mu) \cdot (f - \frac{1}{|V|_\mu} \int_V f d\mu) d\mu$$

$$\leq \sqrt{\int_V \|\Delta_\mu f\|_2^2 d\mu} \sqrt{\int_V \|f - \frac{1}{|V|_\mu} \int_V f d\mu\|_2^2 d\mu}$$

$$\leq \sqrt{\int_V \|\Delta_\mu f\|_2^2 d\mu} \sqrt{C_6 \int_V \|\nabla_\mu f\|_2^2 d\mu}$$

The third Line is from Holder inequality. Therefore we have

$$\int_V \|\nabla_\mu f\|_2^2 d\mu \leq C_6 \int_V \|\Delta_\mu f\|_2^2 d\mu$$

For general dimension $d$, suppose $f = (f_1, \ldots, f_d)^T$, then by Jenson inequality we have

$$\int_V \|\Delta_\mu f\|_2^2 d\mu = \sum_{i \in [n]} \Big( \sum_{1 \leq k \leq d} \sum_{j \in \mathcal{N}_i} \Big(\frac{\omega_{ij}\big(f_k(j) - f_k(i)\big)}{\mu_i}\Big)^2 \Big) \mu_i$$

$$= \sum_{i \in [n]} \Big( \sum_{1 \leq k \leq d} \sum_{j \in \mathcal{N}_i} \Big(\frac{\omega_{ij}\big(f_k(j) - f_k(i)\big)}{\mu_i^1}\Big)^2 \Big) \frac{(\mu_i^1)^2}{\mu_i}$$

$$\leq \sum_{i \in [n]} \Big( \sum_{1 \leq k \leq d} \sum_{j \in \mathcal{N}_i} \frac{\omega_{ij}\big(f_k(j) - f_k(i)\big)^2}{\mu_i^1} \Big) \frac{(\mu_i^1)^2}{\mu_i}$$

$$= \frac{(\mu_i^1)^2}{\mu_i} \int_V \|\nabla_\mu f\|_2^2 d\mu$$

and

$$\int_V \|\Delta_\mu f\|_2^2 d\mu = \sum_{i \in [n]} \Big( \sum_{1 \leq k \leq d} \sum_{j \in \mathcal{N}_i} \Big(\frac{\omega_{ij}\big(f_k(j) - f_k(i)\big)}{\mu_i}\Big)^2 \Big) \mu_i$$

$$\geq \frac{1}{C_6} \sum_{1 \leq k \leq d} \int_V \|\nabla_\mu f_k\|_2^2 d\mu$$

$$= \frac{1}{C_6} \int_V \|\nabla_\mu f\|_2^2 d\mu$$

Thus the proof of (3) is over. When $p = 2$, $\frac{1}{C_6} = \lambda_1(-\Delta_\mu)$, therefore $C_2 = \lambda_1(-\Delta_\mu)$. After we wish to prove

$$C_3 \int \|\nabla_\mu \Delta_\mu f\|_p^p d\mu \leq \int \|\Delta_\mu f\|_p^p d\mu \leq C_4 \int \|\nabla_\mu \Delta_\mu f\|_p^p d\mu \tag{4}$$

For the right-hand side of (4), we first notice that

$$\int_V \Delta_\mu f d\mu = \int_V f \Delta_\mu 1 d\mu = 0$$

Hence by Poincare inequality, we have

$$\int \|\Delta_\mu f\|_p^p d\mu = \int \|\Delta_\mu f - \frac{1}{|V|_\mu} \int_V \Delta_\mu f d\mu\|_p^p d\mu$$

$$\leq C_6 \int \|\nabla_\mu \Delta_\mu f\|_p^p d\mu$$

For the left-hand side of (4), we notice that for any vector-valued function $g$, we have

$$\int \|\nabla_\mu g\|_2^2 d\mu = \sum_{i \in [n]} \sum_{j \in \mathcal{N}_i} \frac{\omega_{ij}}{2} \|g(j) - g(i)\|_2^2$$

$$\leq \sum_{i \in [n]} \sum_{j \in \mathcal{N}_i} \omega_{ij}(\|g(j)\|_2^2 + \|g(i)\|_2^2)$$

$$= 2 \sum_{i \in [n]} (\frac{\sum_{j \in \mathcal{N}_i} \omega_{ij}}{\mu_i}) \|g(i)\|_2^2 \mu_i$$

$$\leq 2M_{max} \int_V \|g\|_2^2 d\mu$$

Replace $g$ by $\Delta_\mu f$ we have the left-hand side of (4). Now suppose $G = (V, E, \omega, \mu)$ is not connected and has connected components $V_1, \ldots, V_l$, then for the right-hand side of (3) we have

$$\int_V \|\Delta_\mu f\|_p^p d\mu = \sum_{1 \leq k \leq l} \int_{V_k} \|\Delta_\mu f\|_p^p d\mu$$

$$\leq C_1 \sum_{1 \leq k \leq l} \int_{V_k} \|\nabla_\mu f\|_p^p d\mu$$

$$= C_1 \int_V \|\nabla_\mu f\|_p^p d\mu$$

Other cases are similar. In the end we show that $\mathcal{E}_W$ and $\mathcal{E}_D$ are special cases of $\mathcal{E}_m^p$. $\mathcal{E}_W$ is a special cases of $\mathcal{E}_m^p$ such that

$$m = 0, p = 2, \mu_i = 1 \ \forall i \in [n]$$

$\mathcal{E}_D$ is a special cases of $\mathcal{E}_m^p$ such that

$$m = 1, p = 2, \omega_{ij} = 2, \mu_i = 1 \ \forall i \in [n], j \in \mathcal{N}_i$$

### C.5 PROOF OF THEOREM 3.3

(1) Suppose $f$ is a solution to the heat equation with initial condition $f(t = 0) = f_0$, then

$$\frac{\partial \int_V \|\nabla_\mu f\|_2^2 d\mu}{\partial t} = 2 \int_V \nabla_\mu f \cdot \nabla_\mu \partial_t f d\mu$$

$$= -2 \int_V \Delta_\mu f \cdot \partial_t f d\mu$$

$$= -2 \int_V \|\Delta_\mu f\|_2^2 d\mu$$

$$\leq -2\lambda_1(-\Delta_\mu) \int_V \|\nabla_\mu f\|_2^2 d\mu$$

Thus we have

$$\frac{\partial(e^{2\lambda_1(-\Delta_\mu)t} \int_V \|\nabla_\mu f\|_2^2 d\mu)}{\partial t} \leq 0$$

Therefore

$$e^{2\lambda_1(-\Delta_\mu)t}\int_V \|\nabla_\mu f\|_2^2 d\mu|_{t=t} \le (e^{2\lambda_1(-\Delta_\mu)t}\int_V \|\nabla_\mu f\|_2^2 d\mu)|_{t=0}$$

In conclusion, we have

$$\int_V \|\nabla_\mu f\|_2^2 d\mu \le e^{-2\lambda_1(-\Delta_\mu)t}\int_V \|\nabla_\mu f_0\|_2^2 d\mu$$

(2) Suppose $f : V \to \mathbb{R}^d$ is an arbitrary vector-valued function on the graph. Then by the proof of Theorem 2.2, we directly compute that

$$\int_V \|\nabla_\mu Pf\|_2^2 d\mu = \int_V \nabla_\mu Pf \cdot \nabla_\mu Pf d\mu$$

$$= -\int_V \Delta_\mu Pf \cdot Pf d\mu$$

$$= -\int_V \Delta_\mu(\Delta_\mu + I)f \cdot (\Delta_\mu + I)f d\mu$$

$$= -\int_V \Delta_\mu^2 f \cdot \Delta_\mu f d\mu - \int_V \Delta_\mu^2 f \cdot f d\mu - \int_V \|\Delta_\mu f\|_2^2 d\mu - \int_V \Delta_\mu f \cdot f d\mu$$

$$= \int_V \|\nabla_\mu \Delta_\mu f\|_2^2 d\mu - 2\int_V \|\Delta_\mu f\|_2^2 d\mu + \int_V \|\nabla_\mu f\|_2^2 d\mu$$

$$\le \lambda_N(-\Delta_\mu)\int_V \|\Delta_\mu f\|_2^2 d\mu - 2\int_V \|\Delta_\mu f\|_2^2 d\mu + \int_V \|\nabla_\mu f\|_2^2 d\mu$$

$$\le (\lambda_N(-\Delta_\mu) - 2)\lambda_1(-\Delta_\mu)\int_V \|\nabla_\mu f\|_2^2 d\mu + \int_V \|\nabla_\mu f\|_2^2 d\mu$$

$$= \Big(1 + (\lambda_N(-\Delta_\mu) - 2)\lambda_1(-\Delta_\mu)\Big)\int_V \|\nabla_\mu f\|_2^2 d\mu$$

Therefore we have

$$\int_V \|\nabla_\mu P^k f\|_2^2 d\mu \le \Big(1 + (\lambda_N(-\Delta_\mu) - 2)\lambda_1(-\Delta_\mu)\Big)^k \int_V \|\nabla_\mu f\|_2^2 d\mu$$

For $\tilde{A}_{sym}$, we specify $P = \tilde{A}_{rw}, \omega_{ij} = 1, \mu_i = D_i + 1$ and notice that $\tilde{A}_{sym} = \tilde{D}^{\frac{1}{2}}\tilde{A}_{rw}\tilde{D}^{-\frac{1}{2}}$. Therefore we have

$$\int_V \|\nabla_\mu \tilde{D}^{-\frac{1}{2}}\tilde{A}_{sym}^k(\tilde{D}^{\frac{1}{2}}f)\|_2^2 d\mu \le \Big(1 + (\lambda_N(-\tilde{\Delta}_{rw-adj}) - 2)\lambda_1(-\tilde{\Delta}_{rw-adj})\Big)^k \int_V \|\nabla_\mu f\|_2^2 d\mu$$

Replace $f$ by $\tilde{D}^{-\frac{1}{2}}g$ we get desired results.

## C.6 Proof of Theorem 4.1

Given a weighted connected graph $G = (V, E, \omega, \mu)$ and a vector-valued function $f : V \to \mathbb{R}^d$, Suppose that every pair of nodes $i, j$ can be connected by a path that has no more than $r$ edges. First, we suppose $d = 1$, Given node i and node j, suppose node i and node j can be connected by a path with no more than $r$ edges, the path is denoted by $P = [i_0 i_1 \ldots i_k]$ with $i_0 = i$ and $i_k = j$. then we have an estimation

$$|f(i) - f(j)|^p \le \big(|f(i) - f(i_1)| + \cdots + |f(i_k) - f(j)|\big)^p$$

$$= \big(\frac{|f(i) - f(i_1)| + \cdots + |f(i_k) - f(j)|}{k + 1}\big)^p (k + 1)^p$$

$$\le \frac{|f(i) - f(i_1)|^p + \cdots + |f(i_k) - f(j)|^p}{k + 1}(k + 1)^p$$

$$= (k + 1)^{p-1}\big(|f(i) - f(i_1)|^p + \cdots + |f(i_k) - f(j)|^p\big)$$

$$\le r^{p-1}\big(|f(i) - f(i_1)|^p + \cdots + |f(i_k) - f(j)|^p\big)$$

Using this result we directly compute that

$$
\begin{aligned}
\int_V \|f - \frac{1}{|V|_\mu} \int_V f d\mu\|_p^p d\mu &= \sum_{i\in[n]} |\sum_{j\in[n]} \frac{(f(i)-f(j))\mu_j}{|V|_\mu}|^p \mu_i \\
&\leq \sum_{i\in[n]}\sum_{j\in[n]} \frac{(|f(i)-f(j)|^p)\mu_j}{|V|_\mu} \mu_i \\
&\leq \sum_{i\in[n]}\sum_{j\in[n]} \frac{(|f(i)-f(i_1)|^p + \cdots + |f(i_k)-f(j)|^p)r^{p-1}\mu_j}{|V|_\mu} \mu_i \\
&\leq \frac{n^2 r^p \mu_{max}^2}{|V|_\mu} \max_{i\in[n], j\in\mathcal{N}_i} |f(i)-f(j)|^p \\
&\leq \sum_{i\in[n]}\sum_{j\in\mathcal{N}_i} \frac{n^2 r^p \mu_{max}^2}{\omega_{min}|V|_\mu} \omega_{ij}|f(j)-f(i)|^p \\
&= \frac{2n^2 r^p \mu_{max}^2}{\omega_{min}|V|_\mu} \int_V \|\nabla_\mu f\|_p^p d\mu
\end{aligned}
$$

The second line is from Jenson inequality. If $p=2$, Suppose $0 = \tilde{\lambda}_0 \leq \cdots \leq \tilde{\lambda}_{n-1}$ are eigenvalues and $v_0, \ldots, v_{n-1}$ are eigenvectors corresponding to eigenvalues respectively such that $\int_V \|v_k\|_2^2 d\mu = 1$, $\forall 1 \leq k \leq n$ and $\int_V v_i \cdot v_j d\mu = 0$, $\forall i \neq j$. Since the graph is connected, the multiplicity of 0 is 1. Then we have

$$
\begin{aligned}
\int_V \|\nabla_\mu f\|_2^2 d\mu &= -\int_V \Delta_\mu f \cdot f d\mu \\
&= \sum_{1\leq k\leq N} \tilde{\lambda}_k c_k^2 v_k \\
&\geq \tilde{\lambda}_1 \sum_{1\leq k\leq N} c_k^2 v_k \\
&= \tilde{\lambda}_1 \int_V \|f - \frac{1}{|V|_\mu}\int_V f d\mu\|_2^2 d\mu
\end{aligned}
$$

Thus when $p=2$, $\frac{1}{C_6} = \tilde{\lambda}_1 = \lambda_1$ For general dimension $d$ we have

$$
\begin{aligned}
&\int_V \|f - \frac{1}{|V|_\mu}\int_V f d\mu\|_p^p d\mu \\
&= \int_V (\|f_1 - \frac{1}{|V|_\mu}\int_V f_1 d\mu\|_p^p + \cdots + \|f_d - \frac{1}{|V|_\mu}\int_V f_d d\mu\|_p^p) d\mu \\
&\leq C_6 (\int_V \|\nabla_\mu f_1\|_p^p + \cdots + \|\nabla_\mu f_d\|_p^p d\mu) \\
&= C_6 \int_V \|\nabla_\mu f\|_p^p d\mu
\end{aligned}
$$

Next, we wish to prove

$$
\int_V \|\nabla_\mu f\|_p^p d\mu \leq C_7 \int_V \|f - \frac{1}{|V|_\mu}\int_V f d\mu\|_p^p d\mu
$$

We directly compute that

$$\int_V \|\nabla_\mu g\|_p^p d\mu = \sum_{i \in [n]} \sum_{j \in \mathcal{N}_i} \frac{\omega_{ij}}{2} \|g(j) - g(i)\|_p^p$$

$$= \sum_{i \in [n]} \sum_{j \in \mathcal{N}_i} \frac{\omega_{ij}}{2} \Big( \sum_{1 \leq k \leq d} |g_k(j) - g_k(i)|^p \Big)$$

$$\leq \sum_{i \in [n]} \sum_{j \in \mathcal{N}_i} \omega_{ij} 2^{p-2} \Big( \sum_{1 \leq k \leq d} \big( |g_k(j)|^p + |g_k(i)|^p \big) \Big)$$

$$= 2^{p-1} \sum_{i \in [n]} \Big( \frac{\sum_{j \in \mathcal{N}_i} \omega_{ij}}{\mu_i} \Big) \|g(i)\|_p^p \mu_i$$

$$\leq 2^{p-1} M_{max} \int_V \|g\|_p^p d\mu$$

Replace $g$ by $f - \frac{1}{|V|} \int_V f d\mu$ we have the desired result.

### C.7   PROOF OF THEOREM 4.2

First, we wish to prove that there exist constants $C_8, C_9$ such that we have the following inequalities

$$\mathrm{TPD}_p(\Delta_\mu X) \leq C_8 \int_V \|\Delta_\mu X\|_p^p d\mu \leq C_9 \big( \mathcal{E}_D(X) \big)^{\frac{p}{2}} \tag{5}$$

For the left-hand side of (5), we directly compute

$$\mathrm{TPD}_p(\Delta_\mu X) = \sum_{i,j \in [n]} \|\Delta_\mu X(j) - \Delta_\mu X(i)\|_p^p$$

$$\leq 2^{p-1} \sum_{i,j \in [n]} \big( \|\Delta_\mu X(j)\|_p^p + \|\Delta_\mu X(i)\|_p^p \big)$$

$$\leq \frac{2^p n}{\mu_{min}} \int_V \|\Delta_\mu X\|_p^p d\mu$$

For the right-hand side of (5), by the proof Theorem 3.2, we have

$$\int_V \|\Delta_\mu X\|_p^p d\mu$$

$$\leq C_1 \int_V \|\nabla_\mu X\|_p^p d\mu$$

$$\leq \frac{C_1}{C_{12}^p} \Big( \int_V \|\nabla_\mu X\|_2^2 d\mu \Big)^{\frac{p}{2}}$$

$$= \frac{C_1}{C_{12}^p} \Big( \sum_{i \in [n]} \sum_{j \in \mathcal{N}_i} \frac{\omega_{ij}}{2} \|X(j) - X(i)\|_2^2 \Big)^{\frac{p}{2}}$$

$$\leq \frac{C_1 \omega_{max}^{\frac{p}{2}}}{2^{\frac{p}{2}} C_{12}^p} \Big( \sum_{i \in [n]} \sum_{j \in \mathcal{N}_i} \|X(j) - X(i)\|_2^2 \Big)^{\frac{p}{2}}$$

$$= \frac{C_1 \omega_{max}^{\frac{p}{2}}}{C_{12}^p} \big( \mathcal{E}_D(X) \big)^{\frac{p}{2}}$$

After, we wish to prove that there exist constants $C_{10}, C_{11}$ such that we have the following inequalities

$$\big( \mathcal{E}_D(X) \big)^{\frac{p}{2}} \leq C_{10} \int_V \|\Delta_\mu X\|_p^p d\mu \leq C_{11} \mathrm{TPD}_p(\Delta_\mu X) \tag{6}$$

For the right-hand side of (6), we have

$$\int_V \|\Delta_\mu X\|_p^p d\mu$$

$$\leq C_4 \int_V \|\nabla_\mu \Delta_\mu X\|_p^p d\mu$$

$$= \frac{C_4}{2} \sum_{i\in[n]} \sum_{j\in\mathcal{N}_i} \omega_{ij} \|\Delta_\mu X(j) - \Delta_\mu X(i)\|_p^p$$

$$\leq \frac{C_4 \omega_{max}}{2} \sum_{i\in[n]} \sum_{j\in\mathcal{N}_i} \|\Delta_\mu X(j) - \Delta_\mu X(i)\|_p^p$$

$$\leq \frac{C_4 \omega_{max}}{2} \sum_{i\in[n]} \sum_{j\in[n]} \|\Delta_\mu X(j) - \Delta_\mu X(i)\|_p^p$$

$$= \frac{C_4 \omega_{max}}{2} \text{TPD}_p(\Delta_\mu X)$$

For the left-hand side of (6), we have

$$\mathcal{E}_D(X) = \sum_{i\in[n]} \sum_{j\in\mathcal{N}_i} \|X(j) - X(i)\|_2^2$$

$$\leq \frac{1}{\omega_{min}} \sum_{i\in[n]} \sum_{j\in\mathcal{N}_i} \omega_{ij} \|X(j) - X(i)\|_2^2$$

$$= \frac{2}{\omega_{min}} \int_V \|\nabla_\mu X\|_2^2 d\mu$$

$$\leq \frac{2C_{15}^2}{\omega_{min}} \left( \int_V \|\nabla_\mu X\|_p^p d\mu \right)^{\frac{2}{p}}$$

