# OpenReview forum: "PoincareNorm: Rethinking Over-smoothing beyond Dirichlet energy"
_ICLR.cc/2025/Conference — Submitted to ICLR 2025_

### Official Review · Reviewer_gVds · 2024-10-25

**Soundness:** 2
**Presentation:** 3
**Contribution:** 2
**Rating:** 3
**Confidence:** 3

**Summary:**

In this paper, the authors propose to address oversmoothing in GNNs by extending the classical measure of oversmoothing, the Dirichlet energy, to its higher-order equivalent that takes into account the discrete derivatives of the graph signal. Inspired by PairNorm, a normalization method to fight oversmoothing based on the Dirichlet energy, the authors propose PoincareNorm, the counterpart of Pairnorm for high-order energies. They show equivalency between many introduced measures and previous ones. Experiments compare Poincarenorm with other strategies, and show an improvement in some cases.

**Strengths:**

- an important topic in GNNs
- a clever idea reutilizing existing tools

**Weaknesses:**

My main comment is about the method itself. The authors introduce new metrics, then dedicate almost the entirety of the paper to showing that all these metrics are equivalent, and that PoincareNorm combat oversmoothing precisely because it is equivalent to previous measures. Never (or barely) is it mentioned what precisely distinguishes the proposed method. Why, if every metrics are equivalent, is it important to use the proposed ones and not others? What are their usefulness, in terms of **additional** implicit bias compared to previous metrics? The whole paper is dedicated to showing the similarities between all the metrics/methods (including previous Pairnorm), it seems to me that it should precisely focus on the differences! And show that these differences make it more appropriate to somehow fight oversmoothing. As of now, I don't see the advantages of the proposed methods.

**Questions:**

See above: if everything is equivalent as shown by the authors, what are the advantages of the proposed method?

---

### Official Review · Reviewer_WevK · 2024-10-29

**Soundness:** 2
**Presentation:** 1
**Contribution:** 2
**Rating:** 3
**Confidence:** 3

**Summary:**

The paper is about node classification in graphs using semi-supervised learning. Authors propose a method to overcome over-smoothing caused by over-reliance on the Dirichlet Energy. The proposed method expands on the Dirichlet Energy in that it uses more than first-order terms on which Dirichlet depends. The method proposes to build on higher order derivative terms to formulate an energy function that does not suffer from over-smoothing. Authors propose to use the Poincare norm function for this. Numerical experiments show results.

**Strengths:**

The paper has a nice literature review on graph methods, spectral and others.

**Weaknesses:**

The paper's problem scope if only defined at section 3. Until there it is not clear what the paper is about. It is unclear to me what the connection between the Poincare norm and higher order derivatives is.

It is unclear to me how important oversmoothing is. I can understand that a non-Euclidean geometry may be more suitable for embedding graphs and there is literature on this topic. But the motivation is not what the paper is built on.

Page 7: Poincare, not Ponicare.

**Questions:**

The paper motivated the use of higher order derivatives. Poincare norm is about non-Euclidean geometry. I could not find the connection in the paper. Can you elaborate on this?

is the expression p-PoincareNorm_0 on page 3 a norm? can you demonstrate this?

---

### Official Review · Reviewer_AE1p · 2024-11-03

**Soundness:** 2
**Presentation:** 1
**Contribution:** 2
**Rating:** 3
**Confidence:** 2

**Summary:**

The authors propose new node dissimilarity measures that capture over-smoothing, inspired by the Dirichlet energy. Notably, their method differs from the existing Dirichlet approaches by capturing information of higher-order gradients. The authors provide theoretical analysis on the properties of their method. They also test their method for a series of semi-supervised tasks against existing methods.

**Strengths:**

The authors provide a thorough literature review on the landscape of oversmoothing, ranging from their characterization, works studying their emergence, and proposed methods to combat them. This resulted in a very sound set of baselines to compare the author's methods against in the experimental section.

The authors also conducts a thorough analysis of their proposed method, motivated by the decay rate-based evaluations in prior literature.

On the experimental front, the authors method achieve noticeable improvements against existing baselines in Table 1.

**Weaknesses:**

While the authors have provided a thorough literature review on oversmoothing, the motivation for capturing higher-order derivatives of the features is not clearly stated. In particular, I was unable to extract how the higher-order derivatives of the feature relates to the characterization of oversmoothing. I feel that addressing this disconnect early on (e.g. why only first-order is not enough) will greatly align the audience's focus for the theoretical section to come.

In Section 3.1, the authors open with reference to prior (albeit recent) definitions of node similarity measures and the Dirichlet energy, as well as lifting Definition 3.1 from the Rusch et al. (2023a) paper. At a glance, the distinction between prior and present contributions are unclear. I recommend the authors cite the prior works explicitly in definitions and theorems lifted from prior context. Furthermore, this preliminary discussion also feels more appropriate in a prior works or introductory setting, in service of motivating the proposed method. I believe that the presentation would be much clearer if Section 3.1 is kept to presenting the proposed method formally (while motivating the theorems themselves -- as opposed to motivating the method as a whole).

While the analysis is thorough, many following discussions stop short of clearly motivating the necessity/importance of the result. For example, under Theorem (4.2) the authors write "Therefore to control $\epsilon_D(X)$ and Laplacian energy... we need only to control $\text{TPD}_p(\Delta _\mu X)$" but stop short of clearly motivating why this is significant.

While the experimental results are clearly significant as presented in Table 1, the improvement of the proposed method against GraphNorm and NodeNorm appears relatively marginal (or even suboptimal) in many instances over Table 2 and 3.

**Questions:**

1. What is the motivation behind the relevancy of higher-order derivatives to addressing the problem of oversmoothing? In other words, what is the (precise) connection between higher-order derivatives and the oversmoothing phenomena? I raise this question because I found it was not clearly articulated after many passes of the paper.


2. What is the y-axis on the left plot in Figure 1? Is it the accuracy (as the title may suggest)? If so, it appears that the best accuracy are achieved with less layers. More importantly, the PoincareNorm method only perform better than other methods in a region where all methods perform subpar than their shallow layer counterparts. How can we conclude that the proposed method is addressing the oversmoothing phenomena from these empirical evidence?

---

### Official Review · Reviewer_39hZ · 2024-11-08

**Soundness:** 3
**Presentation:** 3
**Contribution:** 3
**Rating:** 6
**Confidence:** 3

**Summary:**

The paper introduces a regularization method called PoincareNorm to mitigate the over-smoothing phenomenon in Graph Neural Networks (GNNs). By leveraging bounds on Dirichlet energy through the Poincaré inequality, the authors propose a new class of normalization, termed p-PoincareNorm, which generalizes the existing PairNorm technique. Experimental results on node classification tasks across three datasets demonstrate the effectiveness of the proposed approach, showing performance improvements over previous methods.

**Strengths:**

- The paper provides a theoretical foundation for the proposed norm.
- The discussion on the characteristics of various existing GNN normalization methods for addressing over-smoothing is informative.
- Additionally, the presentation is clear, and the manuscript is easy to follow.

**Weaknesses:**

- Further evaluation on large-scale graph datasets is needed to better assess the effectiveness and efficiency of the proposed method. This would help validate its scalability and applicability in real-world scenarios with complex and extensive graph structures.

- Additionally, a more comprehensive overview of over-smoothing mitigation techniques would provide valuable context. A discussion that situates the proposed method within the broader landscape of existing solutions could help readers understand its unique contributions and limitations relative to other approaches.

**Questions:**

- How were the reported numbers in Tables 1, 2, and 3 calculated, given that experiments were conducted with varying numbers of layers for each model (GCN, GAT, SGC)? Presenting results for each layer configuration would help demonstrate performance stability across different depths.

- Numerous theorems are presented in the paper, but it is unclear which are original contributions. Existing results should be properly cited to clarify which findings are novel and which build on prior work.

---

### Meta-Review · Area_Chair_7oVN · 2024-12-21

**Metareview:**

The authors propose PoincareNorm as a novel regularization term for GNN learning. The motivation is to come up with a measure that captures information about higher-order derivatives of features. However, much of the paper is focused on showing equivalence of PoincareNorm and many other known metrics, not why it is superior.

**Additional Comments On Reviewer Discussion:**

The reviews are predominantly negative and the authors did not provide any rebuttal.

---

### Decision · Program_Chairs · 2025-01-22

Reject